# KRAS, A Prime Mediator in Pancreatic Lipid Synthesis through Extra Mitochondrial Glutamine and Citrate Metabolism

**DOI:** 10.3390/ijms22105070

**Published:** 2021-05-11

**Authors:** Isaac James Muyinda, Jae-Gwang Park, Eun-Jung Jang, Byong-Chul Yoo

**Affiliations:** 1Department of Translational Science, Graduate School of Cancer Science and Policy, National Cancer Center, Goyang-si 10408, Korea; isaacjamesm@gmail.com (I.J.M.); jang@ncc.re.kr (E.-J.J.); 2Uganda Cancer Institute, Mulago-Kampala 3935, Uganda; 3Department of Translational Science, Research Institute, National Cancer Center, Goyang-si 10408, Korea; jg_park@ncc.re.kr

**Keywords:** pancreatic cancer, metabolomics, lipidomics, glutaminolysis

## Abstract

Kirsten rat sarcoma viral oncogene homolog (KRAS)-driven pancreatic cancer is very lethal, with a five-year survival rate of <9%, irrespective of therapeutic advances. Different treatment modalities including chemotherapy, radiotherapy, and immunotherapy demonstrated only marginal efficacies because of pancreatic tumor specificities. Surgery at the early stage of the disease remains the only curative option, although only in 20% of patients with early stage disease. Clinical trials targeting the main oncogenic driver, KRAS, have largely been unsuccessful. Recently, global metabolic reprogramming has been identified in patients with pancreatic cancer and oncogenic KRAS mouse models. The newly reprogrammed metabolic pathways and oncometabolites affect the tumorigenic environment. The development of methods modulating metabolic reprogramming in pancreatic cancer cells might constitute a new approach to its therapy. In this review, we describe the major metabolic pathways providing acetyl-CoA and NADPH essential to sustain lipid synthesis and cell proliferation in pancreatic cancer cells.

## 1. Introduction

Pancreatic cancer is one of the most prevalent cancers and a major cause of cancer-related death worldwide according to the GLOBOCAN 2018. It is very lethal, because out of the 458,918 new cases reported, 432,242 (>90%) deaths were registered [1]. The high mortality rate of pancreatic cancer can be attributed to the lack of appropriate diagnosis and treatment. Indeed, most patients only exhibit symptoms at advanced stages of the disease, rendering early detection difficult. These challenges have prompted intense research in the field of detection and management of pancreatic cancer in the last decade. However, the five-year survival rate has not improved and remains very low.

The etiology of pancreatic cancer is complex and involves many factors, including smoking, high alcohol consumption, obesity (with body mass index (BMI) >30 kg/m^2^), dietary factors (consumption of red/processed meat, high-cholesterol food, fried food, and nitrosamine-containing food), occupational exposures (especially to nickel), advanced age (usually >50 years old), family history (increased risk for individuals with one first-degree relative having pancreatic cancer), type 1 and 2 diabetes mellitus, and infections with *Helicobacter*
*pylori* or hepatitis B and C viruses [2]. Genetic predispositions, including variations or mutations in genes such as *BRCA1*, *BRCA2*, *PALB2*, *ATM, CDKN2A*, *APC*, *MLH1*, *MSH2*, *MSH6*, *PMS2*, *PRSS1*, and *STK11* are also implicated in pancreatic cancer [3]. Particularly, mutations in four major genes including Kirsten rat sarcoma viral oncogene homolog (*KRAS*), tumor protein 53 (p53), and *SMAD4* significantly contribute to the development of pancreatic cancer [4,5]. About 90% of the cases are pancreatic adenocarcinoma (PDAC) of exocrine origin, and <5% are pancreatic neuroendocrine tumor (Pan-NET) arising in the pancreatic endocrine tissue [6]. PDAC is an aggressive cancer with a very poor prognosis, with one- and five-year survival rates after diagnosis of ~24% and 9%, respectively [7].

The pancreas lacks a defined stem cell compartment. The acinar cells of the exocrine gland exhibit a high degree of plasticity and are important for pancreatic homeostasis and regeneration [8]. When exposed to stimuli such as tissue damage, inflammation, or stress, these cells undergo trans-differentiation called acinar-to-ductal metaplasia to acquire an epithelial ductal-like phenotype [9,10]. This process increases the susceptibility of the proto-oncogene KRAS to mutations, leading to pancreatic intra-epithelial neoplasias (PanINs). The evolution of PanIN into fully invasive PDAC requires the gradual and sequential occurrence of inactivating mutations in several tumor suppressor genes, especially TP53, as well as the development of a distinctive microenvironment [11,12].

*TP53* mutations occur in 50–75% of cases and are crucial for the development and rapid progression of tumor malignancy. *CDKN2A* mutations are observed in 5–35% of all pancreatic cancer cases. The normal function of CDNK2A is to prevent G1-S phase transition. Although SMAD4 mutations are present in 30–64% of cases, their role in tumorigenesis remains to be established [13].

The KRAS proto-oncogene product is a small GTPase protein that functions as an ON–OFF molecular switch controlling the transfer of signals from membrane receptors to downstream signaling pathways, (Figure 1). Upon activation of receptor tyrosine kinases (RTKs) by extracellular stimuli, the regulatory activity of guanine exchange factors (GEFs) and GTPase-activating proteins (GAPs) mediate the conversion of KRAS protein from an inactive GDP-bound to an active GTP-bound form transiently, and the GTP-bound KRAS is rapidly hydrolyzed back to an inactive state [14]. In PDAC, KRAS mutations most commonly affect the glycine residues at codons 12, 13, and 61 and prevent the modified KRAS from interacting with GAPs. Therefore, GTP stays bound to KRAS, leading to a constitutively active KRAS and resulting in persistent, uncontrolled downstream signaling, mostly in the mitogen-activated protein kinase (MAPK) and phosphoinsotide-3 kinase (PI3K) pathways [15,16]. KRAS signaling through the MAPK–RAF–MEK–ERK pathway activates downstream transcription factors including FOS, JUN, ETS, and MYC, which promote cancer cell proliferation by activating the expression of proliferative genes involved in cell cycle entry, angiogenesis, and survival of the cell [17]. KRAS interacts with P110α in the PI3K–AKT pathway and augments MAPK oncogenic transcription through the activation of NF-kB signaling. It also supports apoptotic evasion by inhibiting the pro-apoptotic enzyme BAD, and stimulates cell growth and metabolism through the mTOR complex [17,18]. Finally, activation of the TIAM1 pathway promotes cancer cell motility and migration through a Rac–Rho- and Rac–PAX-dependent network [17]. This activated KRAS signaling induces cell hyperproliferation, decreased apoptosis, and development of an invasive and metastatic phenotype.

The current management of pancreatic cancer depends on disease stage and tumor resectability, because surgery is the only curative treatment available. For resectable PDAC, surgery followed by adjuvant chemotherapy is the standard therapeutic course, but it only concerns 10–20% of PDAC patients [19]. Most patients are diagnosed late with advanced stages III and IV, characterized by metastatic spread to the nearby blood vessels and nerves, as well as to distant organs due to the gradual onset of nonspecific symptoms including, jaundice, fatigue, pruritus, dark urine, abdominal pain associated with acholic stools due to obstruction in the biliary tree, anorexia, unexplained weight loss, early satiety, dyspepsia, nausea and less common symptoms such as panniculitis and depression [20]. The absence of symptoms until the later stages, the availability of nonspecific diagnostic tests such as cancer antigen 19-9 which often fail to detect early stage disease, location of the pancreas deep into the body, and the size of tumors which are too small to be seen or felt during routine physical examination further complicates the diagnosis. At advanced stages, the traditional cancer treatment modalities, such as surgery, chemotherapy, and radiotherapy, become irrelevant, and are only routinely used for palliative intent. To date, there is no definite cure for pancreatic cancer [21,22,23].

First-line systemic chemotherapy for nonresectable or borderline resectable tumors consists mainly of purine analogs gemcitabine and capecitabine, or pyrimidine analog 5-fluorouracil (5-FU) used independently as monotherapy, in combination as poly-chemotherapy, or in association with other treatment modalities [24,25]. Poly-chemotherapeutic regimens such as FOLFIRINOX (folinic acid, 5-FU, irinotecan, and oxaliplatin) or the combination of gemcitabine with a nanoparticle albumin-bound paclitaxel (nab-paclitaxel) significantly improve overall survival and quality of life compared to monotherapy, but their use in elderly patients or patients with poor performance status is limited due to the very high drug toxicity [26,27,28].

In addition to the drug delivery challenges posed by desmoplasia, the efficacy of chemotherapy is greatly limited by both intrinsic and acquired chemo resistance [29]. Resistance to gemcitabine is caused by an increased expression of ribonucleotide reductase enzyme, which catalyzes the reduction in ribonucleotides, decreases the expression of human equilibrative nucleoside transporter 1 (hENT1), reduces the active uptake of nucleosides into the cytosol [30,31], and increases the expression of integrin-linked kinase and hypoxia-inducible pro-survival factor BCL2 [32,33]. The presence of cancer-associated fibroblasts (CAFs) in the tumor microenvironment scavenge gemcitabine and metabolize it to an active 2′,2′-difluorodeoxycytidine-5′-triphosphate (dFdCTP) which does not cross the CAF membrane, effectively reducing the concentration of the active metabolite reaching the tumor cells [34]. Resistance to 5FU is mediated by an increased expression of the enzymes thymidylate synthase (TS) and dihydropyrimidine dehydrogenase (DPD) which catabolize 5-FU [35,36].

Radiotherapy is used on locally advanced resectable tumors, but PDAC tumors are particularly radio-resistant, and radiotherapy does not prevent tumor progression [37,38,39]. The high-proliferative PDAC and the dense desmoplasia result in a hypoxic microenvironment, reducing the biological effectiveness of irradiation-induced DNA damage which is highly dependent on oxygen [40,41]. Additionally, overexpression of key DNA damage response genes such as RAD51 promotes an accelerated repair of radiation-induced DNA damage and tumor repopulation [42]. Integrin or SMAD signaling have also been implicated in PDAC radio resistance [43,44]. Finally, radiation induces monocyte recruitment to the irradiated region. Monocytes stimulate tumor cell proliferation and neovascularization, thereby counteracting the effects of the therapy [45].

Immunotherapy has also been employed unsuccessfully because pancreatic cancer cells are not very immunogenic, due to their low mutation burden and their location within an abundant desmoplastic stroma, which favors the formation of cold/non-inflamed tumor with few or no tumor infiltrating lymphocytes (TILs) [46]. The desmoplastic stroma lacks vasculature, which creates a physico-chemical barrier against T cell infiltration and efficient drug delivery. The hypovascularization-induced hypoxia favors the progression of desmoplasia by activating pancreatic stellate cells, driving the recruitment of a variety of immune suppressive cells such as CAFs, T regulatory cells (TREG), and myeloid cells, which further block the activation and infiltration of effector T cells, resulting in low levels of intra tumoral CD8+ T cells [47,48,49]. The recruited macrophages adopt an immunosuppressive, pro-angiogenic M2-like state and prevent CD4+ T cell infiltration in the microenvironment [50]. Consequently, the poor response to immunotherapy due to a low effector T cell tumor infiltration correlates with a shorter overall survival [51]. The insignificant improvement in clinical outcome of cetuximab–gemcitabine-based chemotherapy was attributed to a compensatory activation of Integrin β1 signaling [52,53]. However, the EGFR-specific tyrosine kinase inhibitor erlotinib combined with gemcitabine was effective in both wild-type and mutant KRAS, suggesting the inhibition of other tyrosine kinases or presence of a feedback regulatory mechanism between the mutant KRAS and EGFR [54,55,56].

KRAS mutation, being the major driver in PDAC due to its presence in 90–95% of all pancreatic cancer samples, plays a central role to the KRAS pathway in pancreatic cancer development, and is therefore an ideal target for the management of PDAC. However, attempts to silence KRAS and the related genes by RNA interference and small molecule inhibitors remain challenging and have not yielded positive results, with no novel agent currently available [13,57,58]. Given the treatment challenges and the aggressive nature and lethality of PDAC, further research for new local and systemic therapies focusing on other aspects of the KRAS signaling including its metabolism ought to be exploited. It might provide a whole new approach to PDAC management. It might also improve the existing modalities and potentially enhance the clinical outcomes and overall survival. This review focuses on the mechanism of KRAS-mediated metabolic rewiring. The mechanisms by which different pathways contribute to the enhanced general lipid metabolism observed in pancreatic cancer are emphasized, and the potential targets for cancer control are highlighted.

## 2. Mutant KRAS Signaling and Metabolism

KRAS signaling coordinates numerous metabolic pathways including lipid, nucleotide, and glycolytic pathways to support cellular bioenergetic needs and provide the building blocks necessary for cell proliferation (Figure 2). KRAS mediates the metabolic rewiring mainly via the hypoxia-inducible factor 1 alpha (HIF-1α) and c-myelocytomatosis oncogene product (c-MYC) transcription factors, both of which are downstream of the PI3K/AKT and MAPK pathways, respectively (Figure 2).

The activity of HIF-1α largely depends on its stability, which is defined by the oxygen level. HIF-1α is regulated by various post-translational modifications including hydroxylation, acetylation, and phosphorylation, which are responsible for its interaction with several proteins such as PHD, pVHL (von Hippel–Lindau tumor suppressor gene product), ARD-1, and p300/CBP. Under normoxia, the hydroxylation of proline residues and the acetylation of lysine residues within the oxygen-dependent degradation domain of HIF-1α trigger its binding to pVHL. This causes the pVHL-mediated ubiquitination of HIF-1α and its rapid degradation via the ubiquitin–proteasome pathway [59,60,61]. On the contrary, the absence of oxygen stabilizes the HIF-1α subunit that enters the nucleus, dimerizes with HIF-1β, and interacts with the co-activator p300/CBP to modulate the transcription of numerous hypoxia-inducible target genes, especially those involved in angiogenesis, cell proliferation, survival, and metabolism [59].

In pancreatic cancer, HIF-1α is expressed at the same level in both the hypoxic and the well-oxygenated parts, implying the existence of alternative stability mediators in the oxygenated part [62]. Growth factors and cytokines including platelet-derived growth factor (PDGF), insulin-like growth factor, epidermal growth factor (EGF), transforming growth factor, tumor necrosis factor-α, and interleukin-1β indirectly regulate HIF-1α stabilization under normoxic conditions [63]. This regulation occurs through the PI3K/AKT and MAPK signaling pathways. The RAS/MAPK pathway activated by extracellular signals regulates HIF-1α through phosphorylation. The expression of HIF-1α is enhanced when the PI3K/AKT pathway is activated via RTKs. The mechanisms involved in the regulation of HIF-1α protein levels by PI3K/Akt are unclear, but HIF-1α synthesis is increased upon PI3K/AKT/mTOR induction in the normoxic environment while its degradation is not affected [64]. It is proposed that phosphorylation weakens the interaction of HIF-1α with pVHL, preventing HIF-1α degradation during normoxia [62]. Hypoxia effects on HIF-1α and HIF-1 DNA binding activity are independent of PI3K and MTOR activities.

## 3. Glucose Metabolism

HIF-1α enhances glucose metabolism by increasing the expression of glycolytic enzymes (ALDOA, enolase1, lactate dehydrogenase A, phosphofructokinase, PGK1, and Hexokinase 2). It also promotes glucose uptake by increasing the production of glucose transporter-1 (GLUT1), monocarboxylate transporter 4 (MCT4), and lactate, which facilitates the survival in limited nutrient conditions [65,66]. Most of the glycolytic genes also contain MYC response elements, which also allow MYC protein to synergistically regulate glucose metabolism [62]. HIF-1α and c-myc also reduce the mitochondrial complex IV activity, causing glucose to be shunted away from the tricarboxylic acid (TCA) cycle, which plays a role in supporting the glycolysis, the hexosamine biosynthesis, and nucleotide biosynthesis via the pentose phosphate pathway [67,68,69]. RAS/MAPK-induced MYC specifically drives nucleotide metabolism by diverting the glycolytic intermediates into the pentose phosphate pathway through the increased expression of associated enzymes [70]. PI3K/AKT strengthens the glycolysis by regulating fructose 2,6-bisphosphatase (PFKFB2) expression through the phosphorylation of serine 466 and 483 residues in the C-terminal domain [70]. All these changes result in a high-rate glycolytic pathway, culminating in the formation of pyruvate. The transport of pyruvate into the mitochondrial matrix by its specific carrier is blocked by the overexpression of lactate dehydrogenase (LDH), which preferentially converts pyruvate to lactic acid. HIFα induces the expression of pyruvate dehydrogenase kinase 1 (PDHK1), which inhibits the pyruvate dehydrogenase complex through phosphorylation and blocks the formation of acetyl-CoA. As a consequence, glycolysis-mediated oxidative phosphorylation is reduced [71]. Lactate is pumped out of the cell through the MCT4, which not only prevents its damaging effects, but also prevents the negative feedback caused by the accumulation of pyruvate, thereby further enhancing glycolysis.

## 4. Glutamine Metabolism

KRAS enhances glutamine metabolism and phosphoserine biosynthesis by upregulation of the respective biosynthetic enzymes [72]. The displacement of glucose into anabolic pathways and the excessive conversion of pyruvate to lactic acid saturates the TCA cycle. As a result, cancer cells use glutamine in the anapleurotic reaction to replenish the TCA intermediates [73]. Cells rely on glutamine as an alternative biomolecule for ATP generation and for macromolecular biosynthesis. However, under hypoxia conditions and through KRAS signaling, glutamine is principally used to synthesize lipids de novo [74,75]. KRAS, through the master regulator of metabolism, MYC, induces the expression of the glutamine transporters SLC1A5 and SLC7A5/SLC3A and glutaminase enzymes (GLS). As a consequence, it increases the uptake and conversion of glutamine to glutamate, which in the canonical pathway generates α-ketoglutarate to feed into the TCA cycle [76,77,78]. Glutamine is also redirected to support NADPH production, which is essential to maintain the cellular redox balance and lipid neosynthesis. Indeed, upregulation of the transcription of glutamic-oxaloacetic transaminase 1 (GOT1) and the suppression of glutamate dehydrogenase 1 (GLUD1) [79] shift the glutamine metabolism from the mitochondrial canonical pathway to the non-canonical pathway for NADPH generation [80].

Glutamine is a transporter of nitrogen essential for the synthesis of other amino acids and nucleotides. It also drives the uptake of essential amino acids [81,82]. Its high abundance in the body is due to its de novo biosynthesis mediated by the glutamine synthetase, which catalyzes the condensation of ammonia and glutamate mainly in muscles and lungs. However, cancer cells also show increased uptake from dietary sources, suggesting that glutamine is essential to tumors [83,84,85,86]. Glutamine synthetase is also elevated in a number of cancers, including pancreatic cancer [82].

GLS-generated glutamate is catalyzed in mitochondria into α-ketoglutarate and aspartate by GOT2/AST2. Aspartate is then shuttled to the cytoplasm for further conversion to oxaloacetate (OAA) by GOT1 (cytosolic aspartate aminotransferase 1), then to pyruvate by malate dehydrogenase 1 and malic enzyme 1 (ME1), releasing NADPH [87]. α-Ketoglutarate (α-KG) is a substrate for either decarboxylation by α-ketoglutarate dehydrogenase (α-KGDH) or for reductive carboxylation by isocitrate dehydrogenase 2 (IDH2) [73]. HIF-1α reduces the activity α-KGDH, which normally catalyzes α-KG to succinate, resulting in a metabolic shift to IDH-mediated metabolism to facilitate FA synthesis [88]. The α-ketoglutarate in the TCA cycle undergoes reductive carboxylation to produce citrate. Therefore, glutamine is the major source of citrate and a precursor of lipid production [75,89]. The citrate is shuttled to the cytoplasm through the citrate carriers (CiC/SLC25A1) and, together with the citrate provided by other sources including the diet, is converted to acetyl-CoA by the ATP citrate lyase (ACLY) or the short chain acetyl-CoA synthetase (ACSS2) using ATP [90]. The ACSS2 enzyme is upregulated by hypoxia and poor nutrition conditions and promotes acetate uptake for lipid synthesis in pancreatic cancer cells [91,92].

Non-canonical glutaminolysis, where glutamine is metabolized directly in the cytoplasm via GOT1 and maleate dehydrogenase 1(MDH1) in order to produce pyruvate, is the major source of NADPH for cancer cells to sustain fatty acid synthesis [80]. The inhibition of MDH1 suppresses glutamine metabolism as well as pancreatic cancer growth [93]. The pyruvate generated as an end-product of glutaminolysis is converted to lactic acid by the lactate dehydrogenase enzyme. In this pathway, a key enzyme, isocitrate dehydrogenase 1 (IDH1), catalyzes the conversion of α-ketoglutarate to isocitrate and then to citrate for use in fatty acid synthesis [94,95]. The non-canonical pathway is more vital for PDAC than for normal cells because it is the major source of NADPH. It maintains the redox homeostasis and supports tumor growth. Indeed, oncogenic KRAS drives the glycolysis intermediate metabolites to the non-oxidative arm at the expense of the NADPH-generating oxidative arm, partly through the upregulation of PPP enzymes including ribulose-5-phosphate isomerase and ribulose-5-phosphate-3-epimerase [12,96]. The biosynthesis processes require more NADPH than the redox hemostasis, making NADPH generation a limiting factor for cellular proliferation and growth. The KRAS-mediated bypass of the NADPH-generating oxidative arm allows the occurrence of an alternative mechanism for NADPH production [97]. The knock-down of glucose-6-phosphate dehydrogenase (G6PD) in the oxidative PPP or IDH1 does not affect the NADPH/NADP ratio, while the knock-down of ME1 or GOT1 increases the oxidized:reduced NADP ratio [97]. KRAS-mediated metabolic reprograming of glutamine metabolism contributes to the acquired resistance to chemotherapy, especially against platinum-based drugs. Indeed, GOT1 knock-down in cells resistant to cisplatin reduces their proliferation [98]. Glutaminolysis also decreases cell stress through glutathione (GSH) production, generating alanine and ammonia as catabolic products [99]. KRAS-driven pancreatic cancer cells rely mainly on glutamine for their ability to survive in low glucose and amino acid environments, and are unable to survive glutamine deprivation even in the presence of other nutrients [100,101,102].

Cytosolic IDH1 catalyzes the reversible conversion of isocitrate and α-ketoglutarate, in which NADP is oxidized when isocitrate is synthesized and reduced upon α-ketoglutarate formation [103]. In pancreatic cancer, the high levels of NADPH from the non-canonical glutaminolysis pathway favors the formation of isocitrate and the subsequent lipid synthesis. However, a mutant form of IDH1, containing a heterozygous missense substitution of arginine 132 by histidine, serine, cysteine, or glycine, has been identified in some pancreatic cancer cases. Mutant IDH1 catalyzes a reductive conversion of α-ketoglutarate to 2-hydroxyglutarate [104]. This metabolite promotes DNA and histone methylation that causes cancer cell dedifferentiation. These tumors are more vascularized and more sensitive to both harsh metabolic conditions and chemotherapy compared to IDH1 wild-type tumors. Therefore, the prognosis is more favorable and might explain their very rare occurrence in pancreatic cancer [104].

## 5. Fatty Acid Metabolism

In general, non-dividing cells have a suppressed de novo lipogenesis, with the exception of some cell types, such as adipocytes, hepatocytes, hormone-sensitive cells, and fetal lung cells [105]. Exogenous lipids can sufficiently stimulate cell proliferation, as demonstrated in pancreatic cancer cell lines and cancer cell xenograft mouse models [106,107]. However, the majority of tumor cells preferentially obtain over 90% of the lipids from the neosynthesis of fatty acids, irrespective of the levels of circulating lipids [91]. The elevated lipogenesis might be an adaptive mechanism of dividing cells. Indeed, the circulating fatty acids are reserved to ATP production directly through β-oxidation or indirectly through hepatic gluconeogenesis, which supplies glucose to the glycolytic pathway to meet the high ATP demand. Nevertheless, continuous de novo lipogenesis provides cells with essential membrane building blocks, lipid signaling molecules, and post-translational protein modifications to support rapid cell proliferation and growth [105]. This process is characterized by an increased expression and activity of lipogenic enzymes, including ACLY, fatty acid synthase (FASN), and acetyl-CoA carboxylase 1 (ACC1) regulated by KRAS [62,102,108,109,110]. Glutamate and citrate also positively regulate the expression of ACC1 [111,112]. Increased levels of lipogenic enzymes, particularly FASN, are poor prognostic markers and predict a low response to chemotherapy, especially gemcitabine [70,113]. Expression of the lipogenic enzymes is regulated by the transcription factor sterol response element-binding protein 1 (SREBP1) controlled by the PI3K/AKT signaling pathway, which is upregulated by oncogenic KRAS in pancreatic cancer [114]. Overexpression of SREBP1 correlates with a poor prognosis and shorter overall survival of pancreatic cancer patients [115]. The growth and survival of cancer cells depends on fatty acid neosynthesis, to such an extent that silencing of the associated genes, particularly FASN and ACC, by chemical inhibition or RNA interference, suppresses their proliferation. It also induces apoptosis of the tumor cells as a result of the accumulation of toxic intermediate metabolites such as malonyl-CoA, while non-malignant cells are not affected [116,117,118]. Therefore, upregulation of the fatty acid biosynthetic pathway is initiated at an early stage and plays an essential role in tumor progression, with mutant KRAS mediating both de novo synthesis and β-oxidation of fatty acids [119].

The de novo synthesis of both fatty acids and cholesterol starts with citrate as the carbon source. Citrate is mainly an intermediate product of glutaminolysis, but is also provided by exogenous sources [120]. Citrate can also be newly synthesized from aldehyde derivatives such as acetyl aldehyde or from amino acid metabolism. Aldehyde dehydrogenase 1A3 is elevated in pancreatic cancer and other tumor types [121,122]. ACLY converts cytosolic citrate to acetyl-CoA and OAA. ACC1 carboxylates acetyl-CoA to produce malonyl-CoA, which is further converted by FASN to palmitic acid [105]. Stearoyl-CoA desaturase expression is increased in pancreatic cancer and is controlled by SREBP1. Stearoyl-CoA desaturase promotes the synthesis of 16-carbon palmitoleate and oleate from palmitoyl-CoA and stearoyl-CoA, respectively [123]. The elongated fatty acids undergo further modifications to generate more complex structures such as phospholipids and sphingolipids, which are the main structural components of cell membranes and organelles. The fatty acids are also the source of steroids, eicosanoids, and the precursor arachadonic acid, which function as signaling molecules in inflammatory and immunological responses [75]. Cancer cells preferentially produce saturated fatty acids that might have a protective effect against oxidative stress and chemotherapy-induced cell death, contributing to increased tumoral aggressiveness [105]. Oncogenic KRAS controls the level of cholesterol, another major component of membranes. Indeed, KRAS upregulates HMG-CoA reductase by the transcription factor sterol response element-binding protein 2 (SREBP2), resulting in an increased cholesterol de novo synthesis. Additionally, the receptor-mediated endocytic uptake of cholesterol is stimulated by KRAS through the upregulation of LDL receptors [124,125].

## 6. Fatty Acid Oxidation

The high-proliferative nature of PDAC dictates a continuous availability of both building blocks and ATP to sustain the anabolic processes. Therefore, KRAS mediates both de novo synthesis and β-oxidation of fatty acids to meet the demand [119]. Fatty acid synthesis involves the conversion of acetyl-CoA to malonyl-CoA, which functions as a two-carbon donor in the de novo fatty acid synthesis. It is also an allosteric inhibitor of the carnitine/palmitoyl shuttle system for fatty acid oxidation [126]. Simultaneous fatty acid synthesis and oxidation involve two distinct enzymes, the cytosolic acetyl-CoA carboxylase 1 (ACC1), and mitochondrial membrane acetyl-CoA carboxylase2 (ACC2). These enzymes both generate malonyl-CoA, but the two metabolites are highly segregated and have different fates due to their distinct location within the cell. The ACC1-generated malonyl-CoA is utilized by FASN for the de novo synthesis of fatty acids in the cytosol. The ACC2-generated malonyl-CoA inhibits the carnitine/palmitoyl-transferase 1 (CPT1) activity at the mitochondrial membrane, preventing the transfer of the fatty acyl into the mitochondria for β-oxidation. As a result, the fatty acid and triglyceride (TG) synthesis is increased by using acetyl-CoA generated from glycolysis [111].

Both ACC genes are regulated by different factors, including diet, nutrition status, and hormones. A fat-free carbohydrate-containing diet induces the expression and activity of both ACC1 and ACC2, whereas starvation or diabetes repress both the expression and activity of the ACC enzymes. Insulin upregulates the ACC1 promoter, while glucagon downregulates it [127]. The transcription factors SREBP1 and SREBP2 regulate the lipogenic genes in response to glucose and insulin downstream of KRAS. The enzymes are also regulated by other physiological factors, including allosteric activation by citrate, inhibition by long-chain saturated fatty acyl-CoA, and covalent modification by cAMP-dependent kinase. The latter phosphorylates ACC1 and AMP-activated kinase (AMPK), which in turn phosphorylates ACC1 and ACC2, reducing their activities [128,129]. When the metabolic fuel ATP is needed, ACC1 and ACC2 phosphorylation occurs, leading to reduced levels of malonyl-CoA, increased generation of ATP by fatty acid oxidation, and decreased ATP consumption by anabolic processes [111]. ACCs are also regulated by MYC factor. The cellular levels of MYC and AMPK are inversely correlated, although they regulate the same metabolic processes. Ultimately, the metabolic role of MYC is to increase ATP to support the anabolism and proliferation of cells, whereas AMPK functions to conserve energy and restore ATP:ADP/AMP balance [130]. AMPK activity is suppressed in PDAC cells, mostly by inhibiting the tumor suppressor LKB1, leading to increased ACC activity [62].

KRAS promotes cell proliferation through MYC by ensuring the availability of building blocks and ATP to support macromolecular biosynthesis [131]. MYC is responsible for regulating the glycolysis and the biogenesis and function of the mitochondria. MYC inhibition results in reduced overall mass, an atrophic structure, reduced activity of the organelle, and a decrease in glycolysis, leading to tumor growth suppression and apoptosis [132]. MYC knock-out cells are unable to generate ATP, even in the presence of adequate energy-generating substrates such as glucose, glutamine, and fatty acids. These cells attempt in vain to compensate the energy deficit by upregulating glycolysis and oxidative phosphorylation through the activation of AMPK, which is an MYC-dependent mechanism. They also upregulate FAO through increased LCFA uptake, transport, and metabolism, which is facilitated by the high expression of ACADVL and ACADM enzymes in mitochondria [130].

In pancreatic cancer, HIFα-mediated inhibition of acetyl-CoA formation from pyruvate implies that the mitochondrial membrane ACC2 is unable to generate malonyl-CoA, which in turn prevents the inhibition of CPT1. This allows a continuous uptake and β-oxidation of fatty acids. It is further supported by the expression of mTOR-driven ACSL3, a member of the acyl-CoA synthetase family. ACSL3 specifically regulates fatty acid metabolism and utilization by promoting fatty acid uptake and oxidation and catalyzing the formation of acyl-CoA [133]. Fatty acid oxidation generates large quantities of ATP, enabling the pancreatic cancer cells to sustain high-energy processes such as anabolic synthesis and cell signaling.

## 7. Other Metabolic Processes

Pancreatic cancer cells require a constant supply of nutrients, which are mainly obtained from endogenous synthesis. In addition, KRAS supplements nutrient synthesis by enhancing nutrient acquisition through the upregulation of various uptake receptors. KRAS also activates a mechanism of micropinocytosis, importing lipids and amino acids from the extracellular environment. These nutrients are then used in various biosynthetic processes such as protein synthesis, membrane formation, and ATP generation [134,135]. As MYC stimulates the synthesis of mitochondria, pancreatic cancer cells also display high basal autophagy which is essential for the recycling and proper functioning of mitochondria. This process generates nucleotides and other molecules such as fatty acids and amino acids, which can be used for anabolic processes, ATP generation, or the biogenesis of new mitochondria [133].

## 8. Conclusions

The survival of pancreatic cancer cells depends mainly on three factors, namely, the availability of fatty acids and nucleotides, which constitute the building blocks, NADPH for the redox control and lipid synthesis, and ATP for powering all the energy-dependent processes within the cell. MYC-mediated shunting of glycolytic intermediates into the non-oxidative pentose pathway can generate the necessary nucleotides. However, glutaminolysis is the main contributor to both the redox control, by generating NADPH and ATP, and the membrane formation, through de novo fatty acid synthesis [70,75,80,89]. De novo lipogenesis and NADPH generation are achieved by the malic pathway, due to the shift of the glutamine and citrate metabolism from the intramitochondrial space to the cytoplasm. The newly generated fatty acids and those coming from the extracellular environment are used for the formation of the plasma and organelle membranes during cell proliferation. They are also involved in the generation of ATP through β-oxidation. Fatty acid oxidation supplements the bioenergetically inefficient glycolytic process. Indeed, because of the inactivation of the oxidative phosphorylation, glycolysis produces two ATP molecules, which does not match the cellular needs. The increased uptake and breakdown of glucose therefore serves to provide intermediates for nucleotide production in the pentose pathway rather than ATP synthesis. It also directly facilitates lipid synthesis and fatty acid oxidation by inhibiting pyruvate entry into the mitochondria. Although pancreatic cells display enhanced acquisition of fatty acids through micropinocytosis, autophagy, and increased expression of lipid receptors, 90% of the lipids are newly synthesized in the form of triacylglycerol, irrespective of the levels of circulating lipids [91]. The extramitochondrial, non-canonical glutamine and citrate metabolism generate acetyl-CoA and NADPH, which are essential to de novo lipid synthesis. The newly produced lipids are used to construct membranes, ensuring survival and proliferation of the KRAS-driven pancreatic cancer cells. They therefore might be potential pharmacological targets to tackle highly lethal pancreatic cancer.

The non-canonical pathway might be directly disrupted by inhibiting factors targeting various components, including glutamine transporters or enzymes involved in the malic pathway and lipid biosynthesis. The perturbation of the non-canonical pathway can also be achieved by linking the glycolytic pathways to the TCA cycle. To this aim, the inhibition of PDHK1 would prevent the phosphorylation and inhibition of the PDH complex. It would also facilitate the conversion of pyruvate to acetyl-CoA, which can be carboxylated by ACC2 to malonyl-CoA. Malonyl-CoA inhibits CPT1, and therefore prevents the mitochondrial uptake and oxidation of fatty acids and blocks the ATP generation and anabolic processes. The entry of acetyl-CoA into the TCA cycle might block the aneuploric effect of glutamine, and as a result might inhibit citrate production through the canonical glutaminolysis. Another approach to altering the non-canonical pathway might lie in the inhibition of lactate dehydrogenase. This method might disrupt both glycolysis and glutamine metabolism and induce the accumulation of the common end-product pyruvate. As a result, pyruvate inhibits both processes through a negative feedback mechanism and therefore has a significant effect on lipid synthesis.

## Figures and Tables

**Figure 1 ijms-22-05070-f001:**
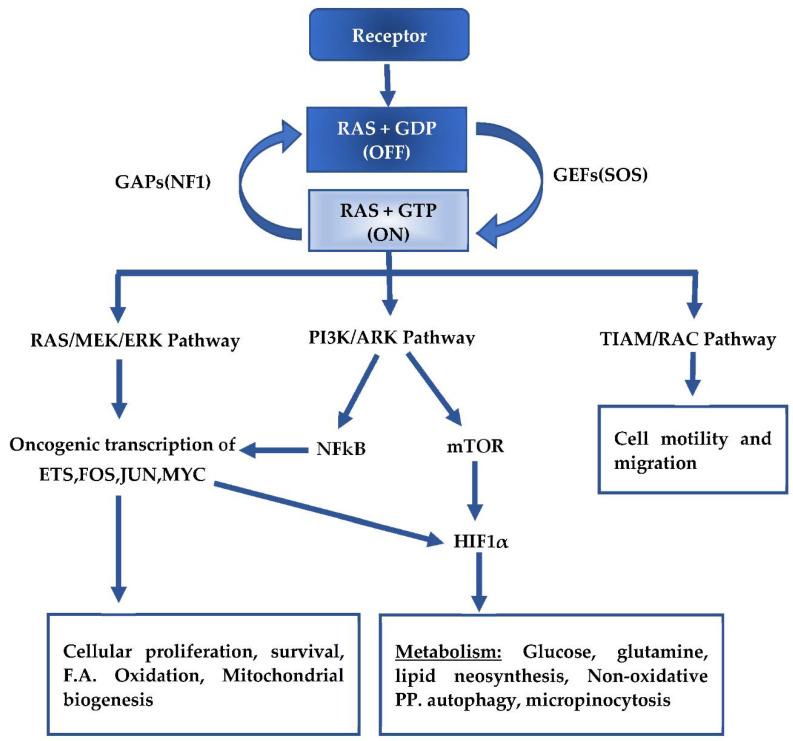
Activation of KRAS and the downstream signaling pathways. “on”—activated KRAS after exchange of GDP for GTP, “off”—inactive KRAS after hydrolysis of GTP to GDP, F.A.—Fatty Acid, PP.—Pentose pathway.

**Figure 2 ijms-22-05070-f002:**
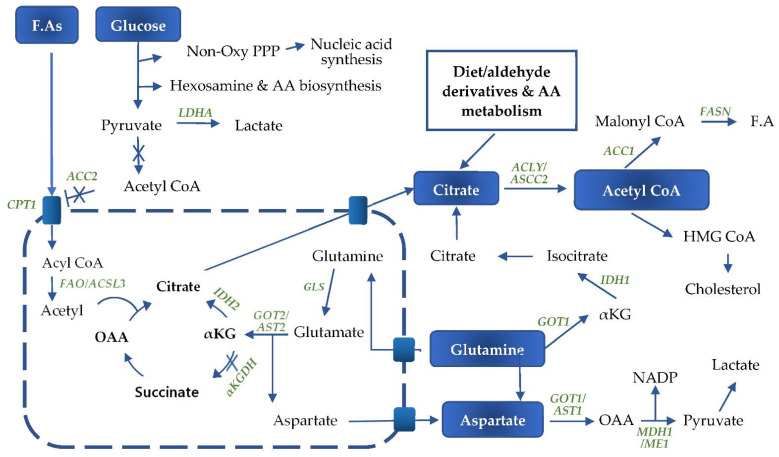
Mutant KRAS induced rewiring of metabolism in pancreatic cancer cells to both the canonical and non-canonical glutaminolysis pathways. Oxy PPP—oxidative pentose phosphate pathway; AA—amino acids. Forward arrow: pathway progression; Forward arrow with a cross: progression is inhibited; Blunt end arrow with a cross: Absence Malonyl-CoA generation from Acetyl-coA prevents normal inhibition of CPT1.

## Data Availability

Not applicable.

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
