# Peer review of "KRAS, A Prime Mediator in Pancreatic Lipid Synthesis through Extra Mitochondrial Glutamine and Citrate Metabolism"

_ijms, 2021, doi:10.3390/ijms22105070_

Round 1
Reviewer 1 Report
The authors presented a comprehensive summary of clinical treatment challenges of KRAS-driven pancreatic cancer, reviewed the mechanisms of KRAS-mediated metabolic rewiring and proposed potential treatment targets.
There are only a few typos that need to be revised, i.e. lines 110, 516, 523,528, 574 etc.
The resolution of the figures needs to be improved.
Author Response
Dear Dr. reviewer
First of all, the authors thank the editorial board for giving us the opportunity to strengthen our manuscript. The manuscript has been carefully revised based upon the comments of reviewers for the publication.
Comment 1 : There are only a few typos that need to be revised, i.e. lines 110, 516, 523,528, 574 etc.
Answer 1 : Several typo errors on pages 3 and 4 were corrected. Besides, we also revised incorrect reference on page 11-19.
Comment 2 : The resolution of the figures needs to be improved.
Answer 2 : We improved the quality of figures to make them easier to recognize than before.
Thank you again for your consideration.
Sincerely Yours,
Byong Chul Yoo

Reviewer 2 Report
In this review article, Isaac James et al. summarized the major mutant pathways and current therapeutic approaches for pancreatic cancer. In second half of the review, they discussed major metabolic pathways that are altered in pancreatic cancer. Overall, this review is thorough and comprehensive. However, several issues need to be addressed.
- Although the title of this review is “Extra mitochondrial glutamine and citrate metabolism major driver of lipid synthesis in pancreatic cancer”, the authors spent nearly two pages introducing therapy of pancreatic cancer. The authors should make it more concise if they would like to include this part in the review.
- In figure 2, the authors could include the upstream genes that regulate the metabolic pathways in pancreatic cancer to make it easier to follow. For example, the authors can label which pathways are regulated by HIF-1a or MYC.
- The authors should consider to change the title because they reviewed not only lipid synthesis, but also glucose, glutamine, and fatty acid oxidation etc.
- The authors may want to review the contribution of p53, CDKN2A, SMAD4 in pancreatic cancer metabolism as well.
- The authors need to correct grammar and spelling. For example, “arguments” should be “augments” on line 85. Line 110, “So, far” should be “So far”.
Author Response
Dear Dr. reviewer
First of all, the authors thank the editorial board for giving us the opportunity to strengthen our manuscript. The manuscript has been carefully revised based upon the comments of reviewers for the publication.
Comment 1 : Although the title of this review is “Extra mitochondrial glutamine and citrate metabolism major driver of lipid synthesis in pancreatic cancer”, the authors spent nearly two pages introducing therapy of pancreatic cancer. The authors should make it more concise if they would like to include this part in the review.
Answer 1 : The purpose of reviewing all the metabolic pathways was to highly their contribution to lipid synthesis therefore we would consider changing the title “Extra mitochondrial glutamine and citrate metabolism major 2 driver of lipid synthesis in pancreatic cancer”
Comment 2 : In figure 2, the authors could include the upstream genes that regulate the metabolic pathways in pancreatic cancer to make it easier to follow. For example, the authors can label which pathways are regulated by HIF-1a or MYC.
Answer 2 : To avoid overcrowding of figure 2, metabolic pathways regulated by either HIF-1 or MYC were summarized in figure 1 in which case it is emphasized that the major regulator of metabolism is HIF-1 being augmented by MYC. The addition of the same information would render figure 1 irrelevant.
Comment 3 : The authors should consider to change the title because they reviewed not only lipid synthesis, but also glucose, glutamine, and fatty acid oxidation etc.
Answer 3 : we changed our title the same as the answer of comment 1.
Comment 4 : The authors may want to review the contribution of p53, CDKN2A, SMAD4 in pancreatic cancer metabolism as well.
Answer 4: Whereas it would be great to review the contribution of p53, CDKN2A, SMAD4 genes in metabolism, this particular review article focused on the contribution of KRAS oncogene in the context of loss of function of the above mentioned genes.
Comment 5 : The authors need to correct grammar and spelling. For example, “arguments” should be “augments” on line 85. Line 110, “So, far” should be “So far”.
Answer 5 : Several typo errors on pages 3 and 4 were corrected. Besides, we also revised incorrect reference on page 11-19.
Thank you again for your consideration.
Sincerely Yours,
Byong Chul Yoo

Round 2
Reviewer 2 Report
The title should be changed if the authors only want to review KRAS in metabolism in pancreatic cancer since the other genes especially p53 also plays critical roles in regulating metabolism.